# Automatic Detection and Identification of Defects by Deep Learning Algorithms from Pulsed Thermography Data

**DOI:** 10.3390/s23094444

**Published:** 2023-05-01

**Authors:** Qiang Fang, Clemente Ibarra-Castanedo, Iván Garrido, Yuxia Duan, Xavier Maldague

**Affiliations:** 1Computer Vision and Systems Laboratory, Department of Electrical and Computer Engineering, Université Laval, 1065, av. de la Médecine, Québec, QC G1V 0A6, Canada; 2GeoTECH Group, Department of Natural Resources and Environmental Engineering, CINTECX, Universidade de Vigo, Campus Universitario de Vigo, 36310 Vigo, Spain; 3School of Physics and Electronics, Central South University, 932 Lushan South Road, Changsha 410083, China

**Keywords:** deep-learning non-destructive evaluation (NDE), automatic defect identification and segmentation, infrared thermography, pulsed thermography, infrared image processing, convolutional neural network

## Abstract

Infrared thermography (IRT), is one of the most interesting techniques to identify different kinds of defects, such as delamination and damage existing for quality management of material. Objective detection and segmentation algorithms in deep learning have been widely applied in image processing, although very rarely in the IRT field. In this paper, spatial deep-learning image processing methods for defect detection and identification were discussed and investigated. The aim in this work is to integrate such deep-learning (DL) models to enable interpretations of thermal images automatically for quality management (QM). That requires achieving a high enough accuracy for each deep-learning method so that they can be used to assist human inspectors based on the training. There are several alternatives of deep Convolutional Neural Networks for detecting the images that were employed in this work. These included: 1. The instance segmentation methods Mask–RCNN (Mask Region-based Convolutional Neural Networks) and Center–Mask; 2. The independent semantic segmentation methods: U-net and Resnet–U-net; 3. The objective localization methods: You Only Look Once (YOLO-v3) and Faster Region-based Convolutional Neural Networks (Fast-er-RCNN). In addition, a regular infrared image segmentation processing combination method (Absolute thermal contrast (ATC) and global threshold) was introduced for comparison. A series of academic samples composed of different materials and containing artificial defects of different shapes and nature (flat-bottom holes, Teflon inserts) were evaluated, and all results were studied to evaluate the efficacy and performance of the proposed algorithms.

## 1. Introduction

Modern industrial production relies heavily on quality management (QM) [1], which is crucial for maintaining high standards in various manufacturing applications, including the aerospace industry. Implementing an efficient QM and control system can provide a significant technological boost to such fields. Structural monitoring is vital for ensuring the health of production lines, and visual inspection systems are increasingly necessary for achieving this goal. However, manual inspection during the quality control stages can be hindered by inspector fatigue, making automatic quality control and defect detection more crucial for improving inspection rates and achieving cost-effective condition monitoring [2].

Among various non-destructive testing (NDT) techniques, infrared thermography (IR) [3] is used to visualize the temperature distribution on the surface of materials, allowing us to “see the un-seen.” Infrared Non-Destructive Evaluation (INDE) aims to identify and categorize Regions of Interest (ROIs) as either defects or anomalies through the analysis of a sequence of images or a single image. Additionally, the objective is to accurately detect these ROIs under consistent conditions.

Pulsed Thermography (PT) [4] is a type of infrared thermography that employs an instantly applied energy impulse to generate and visualize temperature differences on the surface of specimens and detect defects. PT is a faster and more convenient technique compared to other non-destructive techniques and thermography methods. However, PT thermal images can suffer from certain drawbacks, such as edge blurring, non-uniform heating, and low resolution. Moreover, issues related to environmental reflections and specimen geometry can affect the inspection process, leading to a reduced accuracy rate for detecting and evaluating complex internal defects.

When it comes to Nondestructive Evaluation (NDE) for quality management (QM) using pulsed thermography, image processing issues must be addressed to effectively detect defects. Post-infrared image processing has become a crucial aspect of defect detection with infrared thermography to overcome the drawbacks of pulsed thermography. Several state-of-the-art post-image processing methods in pulsed thermography, including Pulsed Phase Thermography (PPT) [5], Principal Component Thermography (PCT) [6], Difference of Absolute Contrast (DAC) [7], Thermographic Signal Reconstruction [8], and Candid Covariance Free Incremental Principal Component Thermography [9], have been documented to yield remarkable results in improving defect visibility during INDE. These methods have been extensively researched and demonstrated to be powerful tools that provide noticeable results in infrared thermography.

Furthermore, conventional machine-learning methods were initially developed to improve feature extraction efficiency, and they have proven to outperform many other approaches in various applications, making them increasingly popular among scientific communities. For instance, (1) Artificial Neural Networks (ANNs) have been used for fiber orientation assessment in composite materials based on interconnected elements (neurons) [10]; (2) Single hidden-layer feedforward neural networks (SLFNs) offer an unlimited number of neurons from a single hidden layer for defect classification [11]; (3) A Support Vector Machine (SVM) automatic classification model has been proposed for breast cancer detection based on thermal pattern images [12]; and (4) the k-means clustering method has been applied to automatic defect detection in fruits for quality classification [13].

The methods mentioned above are pattern-based unsupervised techniques [14], which can extract defect information from temporal and spatial domains. However, there is a need for further research to improve defect visibility and achieve automatic detection in TNDE literature. Additionally, supervised deep learning (DL) [15], a subfield of machine learning, has been shown to perform well in infrared NDT with defect detection, as the vectors extracted from convolutional neural networks can be used as features for detecting defects via Pulsed Thermography. To our knowledge, only a few studies have been conducted in the literature on the automatic detection of infrared thermography using supervised deep-learning algorithms.

Several notable articles have been published on the application of deep-learning algorithms in infrared thermography for automatic defect detection. For instance, in [16], the author proposes the use of deep-learning algorithms to aid in the inspection and evaluation of various infrastructures with water-related problems and thermal bridges. A Mask–RCNN is employed to analyze different results. In [17], the author introduces a deep-learning algorithm to detect abnormalities in temperature patterns and indicate breast pathologies. Abnormal regions can be classified with a high sensitivity rate using a deep convolutional neural network (CNN) with transfer learning. In [18], the author combines deep learning with laser thermography to classify different materials based on thermal signatures. The proposed system works fast, contactless, and with high accuracy. In [19], the author uses a recurrent neural network with an artificial neural network to automatically inspect a non-planar carbon fiber-reinforced plastic sample. The short-term memory can automatically detect defects by analyzing complex data in NDT. The results focus on visualizing specific plates and defect regions at the pixel level. In [20], the author modifies the original Faster–RCNN network through the aggregation of the feature extraction network and adjustment of anchor selection scheme to detect superficial cracks in thermal images. The detection results are evaluated with accuracy and mean average precision, surpassing the original network’s performance. In [21], the author introduces a hybrid of spatial and temporal deep learning for automatic defect detection in infrared thermography. The method effectively reduces uneven illumination and improves contrast between defective and non-defective regions, focusing on regular- and irregular-shaped defects embedded in carbon fiber-reinforced polymer (CFRP).

It is worth noting that previous studies did not thoroughly examine and contrast deep-learning techniques for detection and analysis purposes. Hence, this paper aims to discuss and compare spatial characteristic models utilized in identifying or segmenting defects in infrared thermography. Specifically, three different defect detection methods will be introduced and extensively investigated for the automatic identification and detection of defects using pulsed (infrared) thermography in an infrared system.

The primary objective of this study is to present suitable deep-learning frameworks that can aid in the automatic detection of defects through pulsed thermography for thermal non-destructive evaluations. These frameworks aim to accurately and efficiently extract and separate different types of defects, including less visible cracks, internal defects, and delamination structures, even when the data are limited. To facilitate a comparative analysis of the deep-learning models, a standard infrared segmentation method known as Absolute Thermal Contrast (ATC) [22], with a global threshold, was also introduced.

The key contributions of this research are as follows:A comprehensive and systematic investigation and comparison of three classical deep-learning methods were conducted to analyze the accuracy and efficiency of defect detection using pulsed thermography.An innovative instance segmentation method was introduced to predict the irregular shape of each defect instance in thermal images at the pixel level, enabling efficient defect segmentation and identification for each defect type across different specimens.Experimental modeling and analysis for the post-processing of inspected data based on deep-learning feature extraction techniques have also been introduced.

The structure of this paper is as follows:Section 2 outlines the main principles and methods utilized in this research.Section 3 provides an introduction to pulsed thermography (PT).Section 4 describes the experimental setup, including details on data collection, defective features, and samples.Section 5 presents the spatial deep-learning models used in the investigation, including YOLO-V3 [23], Faster–RCNN [24], U-net [25], Resnet–U-net [26], Mask–RCNN [27], and Center–Mask [28].Section 6 offers a detailed account of the experimental results and training procedures for each method.Section 7 analyzes the results obtained from the experiments.Finally, Section 8 concludes the research and highlights future work in this area.

## 2. Principles

In this section, a detection system trained with pulsed thermography data was proposed to segment and identify defects in thermal images. The spatial characteristic deep-learning model is introduced separately and comparatively in this strategy, as shown in Figure 1. The design of this defect detection system is based on the three types of detection frameworks. The implementation steps can be illustrated as follows:First, the infrared thermal sequences are acquired by the pulsed thermography (PT) system.Secondly, the raw thermal sequences are preprocessed and decomposed by augmentation methods: 1. Principal Component Thermography (PCT), where the sequence decomposes into several orthogonal functions (Empirical Orthogonal Functions: EOF); 2. Flip; 3. Random crop; 4. Shift; 5. Rotation etc.In the final step, the defect regions are recognized via deep neural networks, which visualize the defects with the bounding boxes. All defects must be labeled with the locations, then trained with the deep region neural network.

## 3. Thermography Consideration—Optical Pulsed Thermography

In PT [29,30], a high-power thermal pulse is applied to the surface of the specimen through heat radiation. Due to the heat conduction of the thermal front absorbed by the specimen’s surface, the thermal front travels from the surface and propagates through the materials. As the time elapses, the surface temperature will decrease uniformly for a zone without defects. Conversely, if there is an internal defect beneath the surface (e.g., delamination, disbands, damage, etc.), this defect can become a resistance to heat flow that produces higher temperature patterns at the surface with a decay of temperature, which can be inspected by an infrared (IR) camera. Figure 2 indicates the fundamental principle of pulsed thermography.

In a solid of semi-infinite isotropic conduction, a 1D solution of the propagation of the pulse of a Dirac heat pulse is indicated in Equation (1) as a Fourier mathematical equation,
(1)$$T\left(z,t\right)=T_{0}+\frac{Q}{\sqrt{kpct}}exp\left(-\frac{z^{2}}{4\alpha t}\right)$$
where the energy absorbed by the surface is *Q* $\left[J/m^{2}\right]$ and $T_{0}\left[K\right]$ is the temperature of initialization. The surface temperature progression at *T*(0, *t*) can be written as follows:(2)$$T\left(0,t\right)=T_{0}+\frac{Q}{e\sqrt{\pi t}}$$

From Equation (2), where *e* = *kpc* is effusive. The temperature of the surface evolution following a Dirac heat pulse will decay as a monotonous decrease as $t^{-1/2}$ without defects, while areas with defects will diverge more or less from this behavior based on the actual thermo-physical properties of the region.

## 4. Specimens and Experimental Setting Up

This section may be divided by subheadings. It should provide a concise and precise description of the experimental results, their interpretation, as well as the experimental conclusions that can be drawn.

### 4.1. Experiment Setup

Infrared measurement and the inspected system are the essential parts of collecting infrared data from pulsed thermography. To evaluate the robustness of the proposed algorithms, a certain number of samples were tested. In general, the inspected system used in this experiment consists of: two photographic flash lamps (Balcar FX 60.5 ms thermal pulse) 6.4 kj/flash, an infrared thermal camera, and a personal computer (PC)-Ubuntu 14.04, as shown in Figure 3. To be more detailed, the sampling rate was 157 Hz, a total of three types (steel; CFRP; plexiglass) of eight pieces of specimens were inspected.

The analysis of the thermography process was conducted with the PC (Intel(R) Core (TM) i7-2600 CPU, 3.40 GHz, RAM 16.0 GB, 64-bit, Operating System) and the processing of the thermal data was conducted using the MATLAB computer program R2019a and a Tensor-flow deep-learning open-source library. A mid-wave infrared (MWIR) camera with a special mid-infrared lens (to filter the MWIR spectrum) and two normal lamps were utilized for collecting the infrared data. The normal lamp (containing the entire visible spectrum) was used as an illumination source to illuminate the specimen during the inspection performed inside the laboratory.

### 4.2. Validation Samples Preparation

To evaluate the performance of the proposed method, academic samples were collected independently from three types of materials: plexiglas (Plexi), carbon fiber-reinforced polymer (CFRP), and steel. All the experiments with DL models were conducted under the databases collected from these samples.

As shown in Table 1, the description of eight validation samples in this work is explained. The aspect ratio (size/depth) for all trained and validated samples is designed at (0, 60) to reveal if the detection model has a flexible performance to detect defects. Among the eight validated specimens, the detailed description can be illustrated as follows: The first sample (a) is from plexiglass material with 25 sub-surface circle defects of different diameter and depth.The second sample (b) has eight multiple angle defects that are embedded on the surface of the plexiglass specimen.The third sample (c) is from plexiglass material with 25 sub-surface circle defects of same diameter but different depths, increasing from the left to right column (deeper).The fourth sample (d) plexiglass has 25 circle and quadrilateral defects of various depths and sizes.The fifth sample (e) is a steel sample that has three different diameters of circle defects; the depth being shallower from top to bottom.The sixth sample (f) CFRP has 25 triangle defects embedded in the specimen in the form of a folding plane.The seventh sample (g) CFRP has 25 triangle defects embedded in the specimen in the form of a flat plane.The eighth sample (h) CFRP has 25 triangle defects embedded in the specimen in the form of a curved plane.

### 4.3. Validation Datasets and Features

#### 4.3.1. Acquisition of the Training Database

To maximize the probability of detection, we independently sampled 4000 thermal images in total from the pulsed thermography experiment in three types of materials (plexiglas, carbon fiber-reinforced polymer (CFRP), and steel) to build a training and testing database from pulsed thermography data. As the images used for training should be the same size, the database was split into 512 × 640 pixels.

#### 4.3.2. Calibration of the Data

The marking process was conducted with the two labelling software based on the model type: Colabeler toolkit (YOLO–V3; Faster–RCNN); LabelMe 2.5 toolkit (Mask–RCNN; Center–Mask; U-net; Res–U-net).

Each representative image file from the four types of samples was extracted from the sfmov.format sequence files or matrix raw files. These samples created multiple shapes of defects in the database, such as squares and rectangles.

In the Colabeler toolkit, only one label (square-shape label) was used for all of the different kinds of marks. The bounding boxes were prepared by hand for each of the images, then exported to a .xml file by Colabeler. Each bounded defect was used as training for the algorithm. The process has to be repeated for all images used for training.

In the Labelme toolkit, a different labeling curve from the procedure will be provided regardless of the shape of the defects for segmentation, a labeling curve on each object in the images is then exported to a json.file by Labelme to transform into a large scale object segmentation database (COCO). The elaborate labeling procedure has been explicitly depicted in Figure 4a–c, providing a comprehensive representation of the precise steps involved in the processing of the data.

#### 4.3.3. Preprocessing and Data Augmentation

In the case of the overfitting issue during the training, data augmentation plays a significant role.

We encourage this model to learn the invariant and transformations by using rotation and flipping for the raw images. Since the defects in these materials remain in permanent positions and shapes, they lead to a requirement of capturing images in diverse conditions. As known, the defect is not clear because of the shaping process and/or the specifications of materials that lead to captured images on cluttered background. Those reasons lead to the augmentation of the captured images before entering them into a deep-learning network, which is important. Partial images for the training are undertaken in a preprocessing stage.

We adapted the preprocessed sequence images from feature extraction methods, including Principal Component Thermography (PCT), which extracts meaningful features by dimension reduction and reflects the intuitions of the data. For example, when the data arise from the high dimensional form (sparse and unstable estimation), the PCT can give more redundancy to our classier to enable them to make a better decision.

## 5. Methodologies: Defect Detection Methods by Deep Learning Algorithms

As shown in Figure 5 below, three main deep-learning feature-extraction methods and their implementation steps were introduced: A. Objective localization algorithms: Method 1. Single-stage real-time algorithm-You Only Look Once (YOLO-V3), and Method 2. Two-stage real-time algorithm—Faster Region-based Convolutional Neural Networks—Faster–RCNN; B. Semantic segmentations: Method 3. U-net, and Method 4. Res–U-net; C. Instance segmentation: Method 5. Mask–RCNN and Method 6. Center–Mask; D. Regular thermal segmentation: Method 7. The absolute thermal contrast with global threshold.

### 5.1. Objective Localization Algorithms

Method 1: Real-time defect localization (YOLO-V3)

YOLO-v3 is a proposed supervised deep-learning algorithm that has excellent detection capability both on the large or small objects due to its concatenation involving the merging of the features from the earlier layer with the features from the deeper layer, especially during the infrared nondestructive evaluation with an automatic defect detection task (subsurface defects case).

Processing images with YOLO v3 is quite fast and simple, allowing defects to be detected and localized directly. To perform the feature extraction, residual networks and successive 3 × 3 and 1 × 1 convolutional layers are localized in YOLO-v3 in Figure 6. The skip-connections mechanism was achieved by residual networks through multiple residual units [9,10], which was proposed to improve the performance of object detection, and also solve the gradient vanishing issue. In this research, the YOLO-v3-based deep architecture neural network is proposed to perform the detection of defects (of various sizes). This algorithm includes the implementation of three steps. First, the pictures are resized as the input size. Then, an entire convolutional network is run on these pictures. Lastly, we threshold the detection results based on the model confidence scores.

In Figure 7, an example is shown of an original image (a) and a detected image (b) from the YOLO-V3 network. The CNN was able to distinguish the components, which have a similar thermal pattern with defects during the processing of thermal diffusion, which indicated that the supervised learning method (YOLO-V3) is less influenced by the boundary information in the components.

Method 2: Real-time multiple-stage defect localization model (Faster–RCNN)

Faster–RCNN is a real-time detector that achieved satisfying accuracy with several previous object localization applications in NDT [31]. In 2018, the Faster–RCNN was used for crack detection in an eddy current thermography diagnosis system. The neural network based on a deep architecture was proposed to deal with the problem of accurate crack detection and localization via the preprocessing unsupervised method (Principal Component Analysis).

The deep architecture of Faster–RCNN is composed of several modules (Figure 8):A fully convolutional network, which included five blocks of basic convolutional layers and a Relu layer with a pooling layer to extract feature from the input images.A region proposal network (RPN) connected with the fully convolutional network to obtain the region of interest (RPI).A Fast–RCNN detector using the feature region extracted in the (1)–(2) to achieve bounding box regression and SoftMax classification.

The Faster R–CNN trained from multi-properties, rather than the regular unsupervised method, was limited with respect to certain properties that the defect information contained. An example image detected from Faster–RCNN, as well as a corresponding original thermal image, is shown in Figure 9.

### 5.2. Semantic Defect Segmentation Method

Method 3 defect-segmentation method with U-net network

The U-net is an excellent auto-encoder format model to handle the training data with dimensionality reduction and data augmentation. It is worth evaluating the performance of semantic segmentation by U-net after extracting objective features from the temporal infrared sequence. In the previous article [32], the U-net was employed for the segmentation of wildland and forest fires as a deep-fire convolutional network obtaining very good performance.

The convolutional architecture of U-net is inspired from the auto-encoder network architecture, as indicated in Figure 10. Contracting path maps from the original image to a low dimension vector by extracting meaningful feature representations, and the expansive path reconstructs the output of the desired feature maps. The contracting path is composed of a group of convolutional blocks: convolutional layers; rectified linear unit (ReLU) [33]; and max pooling (dimension reduction). The expansive path included groups of reconstruction blocks to upsample the feature: up-conv (half-reduce the feature channels), concatenation with a feature map from cropping in the contracting path, and so on.

In the final layer, the feature vectors are classified into the target number of the class by 1 × 1 convolution. Moreover, this architecture relies heavily on data augmentation for its performance, which is explained below. The data augmentation strategy from the U-net architecture also brings a significant benefit for the performance for the training. Due to the characteristics of the spatial-thermal temperature sequence, the infrared thermal profile for the defect and non-defect pixels can be distinguished based on the labeling to the force implementation of the supervised learning method (U-net segmentation).

During the cooling period of the thermal data, a temperature change curve over time is obtained on the given image sequence. Therefore, each single thermal frame is fed into this model at the pixel level, and the thermal image can gradually capture the physical properties of temperature variation by U-net. The input values of U-net are thermal temporal evaluation vectors from each pixel. The output label is set either as 1 or 0 corresponding to the defect or non-defect region. During the validation stage, an obtained thermal sequence is selected as the input data after de-background and normalization. The output is a segmented image reconstructed from the predicted value as shown in Figure 11b. Figure 11a is the corresponding original thermal image.

Method 4: Res–U-net for defect semantic segmentation

It is worth investigating comparatively to evaluate thermal sequence databases based on these different defect segmentation methods. As indicated in Figure 12, Res–U-net is an adapted novel encoder/decoder structure evolved from U-net in combination with several structures: residual connections [34]; atrous convolutions [35]; pyramid scene parsing pooling [36]. Res–U-net can infer sequentially the boundary of the objects, the distance transforms of the segmentation mask, the segmentation mask, and a colored reconstruction of the input.

Since residual blocks in Res–U-net can remove vanishing and exploding gradients [37] to a great extent to improve the implementation efficacy of the learning mode and to achieve the pixel level of the segmenting of defects and classification, Res–U-net was compared with other state-of-the-art DL algorithms. The Res–U-net original was performed on the mono-temporal aerial images for the task of semantic segmentation. The framework adapted here for segmenting defects included a Res–U-net framework and a corresponding novel loss function: Dice loss [38]. This reliable framework can perform semantic segmentation, resulting in high-resolution images. To avoid the overfitting, the Res–U-net relied on the data augmentation strategy as well. Each image was rotated to the angle, zoom in/out, flip, and so on to enlarge the datasets of Res–U-net. In Figure 13, a segmented sample from Res–U-net (b) and the corresponding raw images (a) are shown.

### 5.3. Instance Defect Segmentation Algorithm

Method 5: MASK–RCNN for defect segmentation

The Mask–RCNN detection procedure can be considered as either an object detection function or object segmentation function. Compared with the semantic segmentation, the instance segmentation associates each pixel of an image with an instance label. It can forecast a whole segmentation mask for each of those objects and predict which pixels in the input image correspond to each object instance. It also reduces the restriction to the position of defects rather than predicting a group of bounding boxes for the defects. Mask–RCNN is a classical instance segmentation method extended intuitively from Faster–RCN, which is an end-to-end trainable model to achieve pixel-to-pixel alignment segmentation between inputs and outputs of a convolutional backbone architecture. ROI Align preserves spatial orientation of features with no loss of data for extraction over the entire image of the network. This approach efficiently detects objects in an image while simultaneously generating a high-quality segmentation mask for each instance.

Each thermal image was fed into the backbone convolutional network from Mask–RCNN, once some learned region proposal was obtained from the backbone network. These features projected learned region proposals onto convolutional feature maps. Mask–RCNN uses ROI aligning [39] to warp our feature from the convolutional feature map into the right shape then outputs it into two different branches. As shown in Figure 14, there are two different branches providing an output of predicted results. The top branch (blue line box) is a classification score of categories of region proposals and a bounding box for regression of coordinates in the output. In addition, at the bottom (red line box), a segmentation mask is predicted by the model for each of those region proposals to classify for each pixel in that input region proposal whether it is an object. Figure 15 provides an example of an original image from pulsed thermography (a) and a segmented image from Mask–RCNN (b).

Method 6: Central–Mask for defect segmentation 

Since the Mask–RCNN relies on the pre-defined anchors, its influence slowed down for the speed and accuracy in detection. Central–Mask is a simple yet efficient real-time anchor-free instance segmentation. Based on the structure, Central–Mask could be regarded as a novel spatial attention-guided mask (SAG–Mask) branch, adding a free anchor one-stage object detector (FCOS) [40]. A segmentation mask head is located on each detected box with the spatial attention map that helps to aim attention at informative pixels and suppress noise. Figure 16 shows the overview architecture of Center–Mask. A feature pyramid extractor combines with the FCOS box head to predict classification scores and bounding box regression. A spatial attention-guided mask (SAG–MASK) predicts the segmentation map for the defects based on a spatial attention module [41] from each bounding box, which focuses on meaningful pixels and eliminates the noised influence. Central–Mask achieves a faster speed and surprising accuracy better than other state-of-the-art instance segmentation approaches (Mask–RCNN). In this work, we adapted the Central–Mask network for feature extraction and defect segmentation. The main goal is to precisely detect and analyze defect information from the thermal images. The core strategy from this network is to extract the meaningful thermal pattern from the sequence for each specific defect. Figure 17 shows a raw thermal image (a) and a corresponding segmented thermal image (b) from Center–Mask. Each defect is precisely localized and segmented by the Mask.

### 5.4. Regular Infrared Defect Detection Algorithm

Method 7: Absolute thermal contrast (ATC) with global threshold (GT)

In combination with a global threshold method (GT), the ATC was adapted for the procedure of segmenting defects areas. The vital concept of this method was to compare the grey level of the pixel in the image coordinated (x,y) with the average grey level of a sound region of the sample, and it is often adapted in infrared image processing. Equation (3) describes how this method works: where $T_{atc}$ is the grey level in the ATC image in the coordinate (x,y) of the ATC image. $T_{d}\left(x,y\right)$ is the average grey level of the group pixels in the defect region and $T_{s}\left(x,y\right)$ is the average temperature of a nearly sound region.
(3)$$T_{atc}=T_{d}\left(x,y\right)-\;T_{s}\left(x,y\right)$$

Figure 18 provides an example of the segmentation with this method: (a) The raw image from pulsed thermography; and (b) The corresponding segmented image in Method 7. This method made it possible to reduce the effect from non-uniform heating and remove some thermal pattern noises.

## 6. Experimental Results and Implementation Details

### 6.1. Training

The training procedure for deep learning models was set according to the following principles for different neural network architecture parameters adjusted based on the Pytorch framework. The training processing was conducted on a GeForce GTX1080TI about 30 min. The operating system is set as: Ubuntu 16.04. The framework of the learning model is set as: Darknet. CPU: i7-7700k. Memory: 16GB, GPU: NVIDIA GeForce GTX1080TI.

For each modeling training procedure and hyperparameters setting, we configurated the parameters and time speed, which purely ran on the CPU for 1000 thermal images with defects during the training, as shown below in Table 2. As indicated, the Center–Mask runs at a faster speed and spent the lowest time on the CPU for the training of all the thermal images of defect detection compared with other baseline modeling. The U-net network was the slowest speed and took the longest time to process the thermal images for the training of, and obtaining the feature of, the defects. Further, based on the CPU time of the objective-detection methods, it has been proven that it has obvious advantages and a faster speed that outperformed other modelings’. This could be a crucial factor in the industry to choose the optimized modeling for automatic defect detection with infrared-nondestructive evaluation when it considers its training time and procedure.

### 6.2. Evaluation Metrics

F-score and the probability of detection [45] are introduced to analyze the capability of detection of each detection deep-learning model, which is interpreted by Equations (4)–(7). The precision means the ratio from the cases contain the defects over the cases that are recognized by the system that contains the defects, which represent how accurate the system is in identifying the defects. The recall means the system correctly recognized the defects over the cases that actually contained the defects. The precision and recall values heavily depend on the confidences scores that the system is setting. The F-score is a method to estimate the detection and segmentation capability from these algorithms. β is a value to represent the weight between the precision and recall value. In this work, the recall is a metric that is more influential in evaluating the performance. Therefore, β is equal to 2. The POD reveals the accuracy of the method to detect the defects, which are always calculated at a specific confidence score value. Although the POD keeps the same mathematical format as the recall in the equation, POD represents a further explanation in quantifying research with NDT inspectors. In this work, we set the threshold for CTS at 75% for POD metric.
(4)$$\mathrm{Precision}=\frac{\mathrm{TP}}{\mathrm{TP}+\mathrm{FP}}$$
(5)$$\mathrm{Recall}=\frac{\mathrm{TP}}{\mathrm{TP}+\mathrm{FN}}$$
(6)$$\mathrm{POD}=\frac{\mathrm{TP}}{\mathrm{TP}+\mathrm{FN}}\;(\mathrm{CTS}=75\%)$$
(7)$$F\;\mathrm{score}=\left(\beta^{2}+1\right)\frac{\mathrm{Precision}\times \mathrm{Recall}}{\left(\beta^{2}\times \mathrm{Precision}\right)+\mathrm{Recall}}$$
where TP is true positive, and FN is the false negative representing the number of the defects that have not been detected.

Meanwhile, FP is the false positive defect representing the defects that are wrongly detected as defects when they are in fact not defects. Moreover, the confidence threshold score (CTS) was defined as a standard for measuring the accuracy of detecting corresponding objects in each dataset. CTS is a simple measurement standard that can be used for any task that yields a prediction range (bounding boxes, segmented maps) in the output regarding the ground truth.

### 6.3. Learning Curves

In Figure 19a–f, each deep-learning model was trained for 1000 epochs, respectively. Figure 19a shows the average loss curve for training and validation process for the Mask–CNN model. The training-loss curves decreased while the number of iterations increased. The loss significantly decreased during the first 200 batches, then gradually flattened out around 0.225 as the batch number of the iterations increases and then remains steady. The validation loss coverage involves similar loss. This indicates that the performance of Mask–RCNN was promising during the training procedure. In Figure 19b, the loss curves of the Center–Mask model have a similar momentum to that of the Mask–CNN loss but more smoothly. The training loss stably decreased as well, while the whole number of iterations increased and then converged around 0.341.

In comparison with Figure 19a,b, the four other DL models in Figure 19c–f seem to maintain a similar momentum. The average curve of the training loss became more dramatic while oscillating decreased in the first 500 epochs before flattening out late. The loss curves of steel stabilized at a value lower than 0.5 after 500 epochs. As a result, based on the obtained model, the loss from the six different kinds of deep-learning models further indicated an impressive performance during the whole procedure (training and validation) when it was applied on the defect segmentation and localization of composite materials.

### 6.4. Detection Results

This model provided the shape and location of each defect detection results based on the labeled images with ground truth.

In Table 3f, the noise of the input image is the main factor affecting the segmentation results. As indicated in the U-net result, the segmented image is not clear. The segmentation boundary is still blurry. A preprocessed image from principal component analysis (PCA) was added in the validation database to verify whether the segmentation effect will be better after denoising in the Res–U-net model training. From the results, it seems the performance improved to some extent, and the test result of Resnet–U-net gave a better performance than the original U-net.

Table 3 also shows the visualized results from six deep-learning algorithms. Specifically, in Sample (g), the defect feature from the sample indicated clearly that these deep-learning methods show excellent defect detection capability. However, it is obvious that the comparison methods (semantic segmentation) have a substantial shortcoming. These method results are affected by the non-defect area in Sample (g), whereas the Resnet–U-net can be conducted without false detection. Compared with Resnet–U-net, the original U-net is more sensitive to fix patterns—noise and non-uniform heating from thermography due to the higher false detection rates in the result of Sample (g). Therefore, U-net cannot detect the specific thermal data very well because U-net is too insensitive to defect information. Note that the introduced model Res–U-net can ensure correct detection while effectively prohibiting noise interference.

In terms of Sample (d) (steel) and Sample (e) (plexiglass), it is still quite challenging to detect the defects and abnormal areas because the background and noisy information represent a high percentage around the defect information region in the sample. The result of Sample (d) indicated that Faster–RCNN failed in detecting the less visible defects. For the model of Faster–RCNN, although it introduces a hierarchical structure of deep architecture to extract semantic information in the images, there is still a failure to distinguish the boundary noise information from the steel sample. On the other hand, YOLO-V3 is slightly more effective in comparison with a Faster–RCNN based on the detected results on eight evaluation samples (mAP = 0.75 IOU metric). This further illustrates the introduced model; YOLO-V3 leads to good identification accuracy as a single-stage detector in comparison with the other state-of-the-art methods.

For the instance-segmentation method, the segmented images (Center–Mask, Mask–RCNN) show some indistinguishable results from the ground truth. Several types of defects are detected, which include the shapes of a circle, square, and rectangle. Table 4 shows the detection results of the defects by training, using the instance-segmentation model: Mask–RCNN/Center–Mask model. Particularly, since the training database is composed of regular shapes and permanent angles with circle and square shapes, the testing results in Table 4 show that the irregularly distributed defects with multiple angles are detected accurately, which indicated that the Mask–RCNN/Center–Mask spatial detection model can enhance detection performance based on the instance segmentation of pixel-to-pixel alignment.

As a result, it is not enough to only acknowledge the semantic information; it is more impactful to know how to obtain the low semantic information from defects under the interference of objective noisy conditions. In contrast, for the instance-segmentation models, not only did this illustrate a better segmentation performance for the plexiglass samples, but it also has excellent detection capability for the steel and CFRP samples.

To further analyze the robustness of the learning model in comparison with the state-of-the-art ATC, Samples (a)–(h) were further adapted to carry out a detailed analysis. For instance, in Samples (a)–(e) of plexiglass and steel, in comparison with the introduced method from instance and semantic segmentation, the poorer detection of the comparison methods (ATC) is obvious since the semantic information is unclear, and the segmented defects are not obvious. Then, for Samples (f)–(h) from CFRP, which were limited by the accuracy of IR camera, the detection results of absolute thermal contrast (ATC) are still disappointing, whereas for the DL methods, the results are far superior to ATC. Therefore, for the regular- and irregular-shaped specimens, the overall performance of the DL methods is markedly better than all the state-of-the-art methods (ATC and global threshold).

### 6.5. Reliability Assessment Using Probability of Detection (POD)

The reliability assessment metric of subsurface defects detection. The probability of detection [28] has been further assessed to quantify the performance of these six DL models in this task, which can be expressed as a function of aspect ratio through a POD curve. Each curve was plotted in Figure 20 with respect to the aspect ratio (size/depth) to indicate the quantitative analysis for various sizes and depths of defects with deep-learning models. The results of probability of detection (POD) are based on the defect regions detected from the deep neural network methods based on the referenced ground truth.

Figure 20 (CTS = 0.75) indicates the final POD scores obtained from all the samples (a)–(h) for each DL algorithm. The POD of the instance-segmentation method Center–Mask-based approach has a notable performance, and the highest POD scores in comparison to other approaches represent the highest detectability. Then, the staged objective localization methods (YOLO-V3; Fast-RCNN) have a faster and medium detection accuracy. The semantic segmentation method (U-net; Res–U-net) obtained less accuracy due to the fixed pattern noise and non-uniform heating from the infrared thermal data. However, all of these six DL models surpass the state-of-the-art method (absolute thermal contrast: ATC) and function automatically.

The results from different samples for POD validation of methods are indicated in Table 5 (CTS = 0.75). The results are compared using the thermal frames acquired from the pulsed thermography (PT) on each plexiglass/CFRP/steel sample in Section 4.2. In Table 5, the instance-segmentation method (Center–Mask) still shows an acceptable segmented result for all samples provided. Since the instance-segmentation method can capture feature differences over each pixel, the pretrained model (Center–Mask; Mask–RCNN) has the intrinsic capability to segment defects from background information based on a learning and labeling process. For the semantic-segmentation method, as discussed previously, the original U-net model fails to defect the deeper defects region, such as some defect regions of sample (c), since the spatial features from the infrared sequences in defect regions are difficult to obtain due to the non-uniform heating and fixed pattern noise from the boundary region in this case. In contrast, it is worth noting that the introduced method from the instance-segmentation model (Center–Mask) obtained the best performance in Sample (c) in comparison with the other methods.

The results from Samples (a)–(h), analyzed by the POD evaluation metrics, are indicated in Table 5. The performance is compared with thermal images and sequences collected from pulsed thermography experiments on each specimen. On Samples (a)–(d), the YOLO-V3 and Center–Mask show commensurable results, while the other four algorithms (U-net; Res–U-net; Mask–RCNN; Faster–RCNN; ATC) obtain a less accurate performance. Due to the hierarchical structure of deep architecture in neural networks, each DL model has a strong capability to separate low-degree background information from raw thermal sequences (fix-pattern noise; non-uniform heating) and extract defect feature components. However, it is worth mentioning that the semantic method (U-net) model still fails to detect when it encounters a challenging situation (non-uniform heating), just like Sample (e), while the instance segmentation (Center–Mask) maintains high performance in POD evaluation.

Furthermore, the results validated on the total databases from eight representative Samples (a)–(h) are indicated in Table 6. These results in Table 6 indicated that the Center–Mask learning model achieves the best performance comparable to that of the other benchmark detection methods (semantic segmentation; defect localisation). This network may be able to obtain a relatively stable performance due to the fact that it involves a deep architecture to extract the features driven by the tasks and instance masks.

Table 6 indicated the precision, recall, and F-score analysis result from each algorithm (CTS = 0.75). The Pr represents the precision value, and the Re represents the recall value. The F-score of comparison on Specimens (a)–(c) is relatively high for each proposed DL model. However, specifically for Sample (g), the F-score of U-net is 57.2%, which indicates that U-net only partially detects defects due to noise influences. Whereas the instance segmentation method (Mask–RCNN; Center–RCNN) can reach 76.5%, 83% in Sample (g). From Sample (h), the F-score of semantic segmentation (U-net, Res–U-net) can only achieve 71.8%, 77% in several, and the remaining two comparison methods give the following results: Center–Mask (86%) and Yolo-v3 (79%).

The average F-scores for all eight specimens involving the comparison methods from six different DL algorithms (Faster–RCNN, YOLO-V3, U-net, Res–U-net, Mask–RCNN, Center–Mask) provide 72.62%, 79.8%, 67.25%, 73.66%, 74.8%, and 82.55% defect detection capability, respectively. The instance-segmentation method (Center–Mask) gives the highest capability for detection on average. As a result, the instance-segmentation method is relatively better than the semantic-segmentation methods in terms of detection ability due to noise influence from thermography in this circumstance.

In this experiment, the samples from three types of materials are divided into different geometric distributions (regular- and irregular-shape defects). Due to the different geometric architecture of the regular and the irregular defects, this leads to inconsistent thermal diffusion. Correspondingly, this inconsistent thermal diffusion causes the data distribution to be quite different. It can be concluded based on the comparison of the algorithms that it can be difficult for a single DL model to detect all defects effectively. Thus, we adapt the three types of deep-learning algorithms separately training for the different types of samples and also compare the results with other state-of-the-art methods.

### 6.6. Mean–Average Precision (mAP)

Average Precision (AP) [40] is also an indicator that is analysed for the relationship between the precision and recall values. In this section, the mean–average precision (mAP) metric is introduced to further analyze the top-four detection model ranking in POD analysis in Section 6.5 (Center–Mask; Mask–RCNN; YOLO-V3; Faster–RCNN).

The average precision (AP) [46] is calculated based on the indicated bounding boxes and the different confidence thresholding scores from the DL models in object localization and instance segmentation. The total AP and the precision-recall plots obtained when four different deep-learning architectures were adapted are compared in Figure 21 (including all of the confidence-thresholding score values). The detection results show that the Center–Mask still displays the best performance. Taking the 1000 infrared images from the eight representative specimens, defects can be detected, and the recall and precision can be calculated by the confidence scores, as shown in Figure 21 below. The AP of all 1000 thermal images is 75.05%. For instance, in the instance-segmentation model, the AP is calculated as 75.33% and 81.06%, respectively, for Mask–RCNN and Center–Mask (Figure 21a,b). Therefore, the detection performance is better in Center–Mask than in Mask–RCNN. For the object localization model, the AP is calculated as 76.63% and 71.06%, for YOLO-V3 and Faster–RCNN, respectively (Figure 21c,d). Therefore, the detection performance is better in YOLO-V3 than in Faster–RCNN.

### 6.7. Running Time Complexity

Further, frames per second (fps) [35] was introduced as an idea to certify how many images can be processed in a unit (1 s) time by each deep learning model in order to analyze the running time complexity for the model. In Figure 22, the running time complexity of each model was indicated, which illustrates the average time to detect or predict a defect in each frame (picture) from a DL model: the higher the value in the graph, the faster speed the DL algorithm has.

Based on the analysis of Figure 22, the objective localization approach significantly achieved the fastest speed among the models. For the instance-segmentation detector, Center–Mask has increased the time per frame from the state-of-the-art method ATC: 0.5 fps to 12 fps. Then, Mask–RCNN also achieved a time per frame of 5 fps, which increased the processing speed significantly in comparison with the regular thermal threshold segmentation method (ATC) in thermography.

Moreover, we further analyzed the speed rate in comparison with other state-of-the-art networks and YOLO-V3 still obtained the fastest running time speed to process images due to the reason that it is a one-stage real-time detector and has a much faster speed than other detectors (such as Mask–RCNN; Faster–RCNN). The RCNN methods are relatively slow since these models are two-stage procedures (Region Proposal Network (RPN); ROI pooling). Whereas, as indicated previously the POD curves in Figure 20 (Section 6.5), Center–Mask still achieved the highest POD scores during the whole validation process based on the different aspect ratio values (size/depth). Therefore, in this work, Center–Mask is the most promising to obtain the highest accuracy but YOLO-V3 is the most efficient, which has the faster time frame speed.

## 7. Results Analysis

The deep segmentation models gave attractive results for the Plexiglass/CFRP/Steel materials defects identification evaluation. This project focused on building and fine-tuning the training parameters for those defects. To improve the accuracy of the detection model, the way the dataset is built has a significant impact.

According to the results obtained, the following analyses and points of this experiment were concluded below:To implement a robust detection model, the databases must include enough samples. One way to effectively improve is to increase the size of the dataset by including multiscale images. A database composed of images on different scales (larger or smaller), enables the training to be sensitive to those new dimensions. This would increase the robustness of the deep segmentation algorithms facing larger defects, as well as improve the results on blurry pictures. To help reduce false alarms in the algorithm results and be more convenient for the user, implementing different types of labels is necessary. In the case of this project, each section was labeled with a defect in the spatial segmentation training (Mask–RCNN; U-net; Res–U-net). The proposal is to add different classifications. For example, including the name of the shape of the defect: circle, triangle, or some false positive cases (lighting spots, scratches) would be beneficial. This would allow the algorithm to not detect these shapes as a defect, and, thus, reduce the number of false alarms.Another critical point in this experiment to be considered is the marking process. In comparison to other objective detection methods, Mask–RCNN/Center–Mask especially involves a pixel-based marking approach that could mark the defects accurately, as opposed to marking a considerable area around each defect. It can rapidly and easily annotate the object without the bounding boxes restrictions in most cases. In comparison with an instance-segmentation method, U-net and Res–U-net are the auto-encoder format DL models that can be trained based on each pixel level to semantically segment defect pixels from sound pixels. However due to the burden of tackling massive temporal data of thermal frames, U-net and Res–U-net have less time efficiency and high time complexity on the thermal data in comparison to the instance-segmentation model. Therefore, building and creating more diverse and representative training samples is the key point in the future work in this research. There are several ways in which the size of the dataset can be effectively increased. Through data augmentation involving rotation, horizon flipping, and vertical shifts, the deep neural network model could learn the transformations further. By having different scales of larger or smaller training images, the learning procedure will be more sensitive to those new dimensions. This would also enhance the robustness of the algorithm to train for the detection of large defects and improve the results of grayscale images.In addition, the specific training gave results for specific defects in the academic samples. In this work, training only involved using square, circle, and rectangle defects of plexiglass, CFRP, and steel samples. The detection results indicate that similar defects could be detected on other types of training samples. However, the results also show that if the learning model is tested on other defects that the model did not learn on, it would not be an accurate system to rely on. Hence, to use the deep-learning algorithm for training, we should clearly define the type of sample we are working on and enlarge the robustness of the system to learn this type of sample during the neural network training procedure. In addition, due to the time limitation, we simply labeled all the visible defects of each sample in this experiment. However, if we want to extract the feature map completely for each defect area, the positioning of less visible defects in infrared data will be a significant but challenging issue in further research.A specific limitation of the objective localization algorithms is the influence of the labeling process. Although fast and efficient to use, the bounding boxes also led to some restrictions in most cases. As can be seen, when the circle is present in bounding box, this involves a defect that is totally bounded by the box. However, this shows that although the entire defect is contained, the bounding box also extracted the non-defect area, which possibly introduces multiple errors and less accuracy in the results. The proposal is to make a pixel-based labeling to achieve integrity in the image segmentation, which would only label the defects and not a considerable area around each defect. This proposition can be further clarified by segmentation methods. The results presented here lead to a more reliable defects characterization with pulsed thermography (PT).A good defect characterization is essential to not replace parts that could yet be used and to not leave critically damaged components without the needed repair. Therefore, these results are important, especially, e.g., in the designing of autonomous diagnosis NDT systems, which can make decisions regarding the integrity of the inspected part by themselves. In this work, three different types of automatic detection, being intelligent techniques, to combine with infrared thermography could improve the detection with industrial applications based on each group of results in the previous section. The critically damaged components could be easier identified and maintained the component that could be used by those algorithms with a high AP rate (81.06%). However, the instance segmentation (e.g., Center–Mask) provided the highest detection rate associated with vivid segmentation results among three different algorithms to provides the better solution of detection capability compared with the conventional thermal inspection method in industries. Therefore, it could be able to apply and contribute to current industrialized infrared inspection and controlling system.Future work includes: (a) Tests that can be performed with the instance segmentation method and other NDT techniques based on images like stereography and holography; (b) The best technique, method instance-segmentation method (Center–Mask), which can still be improved by tuning the network parameters; (c) Since the CNN technique achieves excellent performance, other network architectures must be tested and compared in the future to specify the best intelligent tool for defect measurement with infrared images.

## 8. Conclusions

In this work, six spatial deep-learning models, involving instance segmentation (Mask–RCNN; Center–Mask), autoencoder format semantic segmentation (U-net; Res–U-net), and the object localization model (YOLO-V3; Faster–RCNN) are applied for defect detection in infrared thermography. The evaluated results and analysis from different geometric specimens of plexiglass, CFRP, and steel specimen with different aspect ratios (size/depth) are indicated in Section 6. Each POD curve is related to the defect sizes that assess the quality of the results to land smoothly in the case of catastrophic failure results. These spatial deep-learning models are separately and comparatively discussed in brief. Future work will focus on the detection of more complicated structured materials through the modification and combination of different spatial and transient deep-learning models.

## Figures and Tables

**Figure 1 sensors-23-04444-f001:**
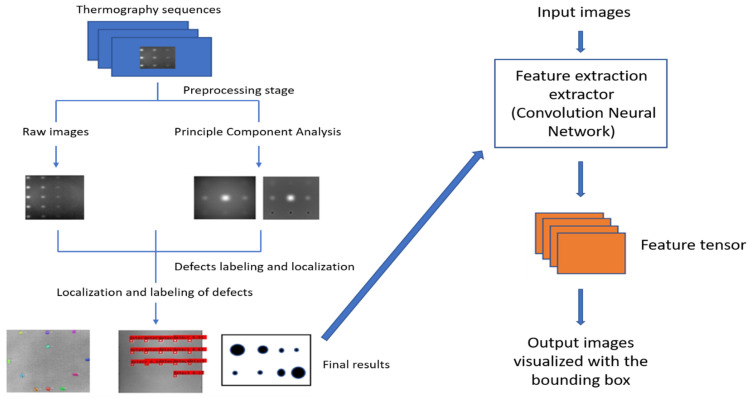
Proposed detection strategy (Instance segmentation; Semantic segmentation; objective detection).

**Figure 2 sensors-23-04444-f002:**
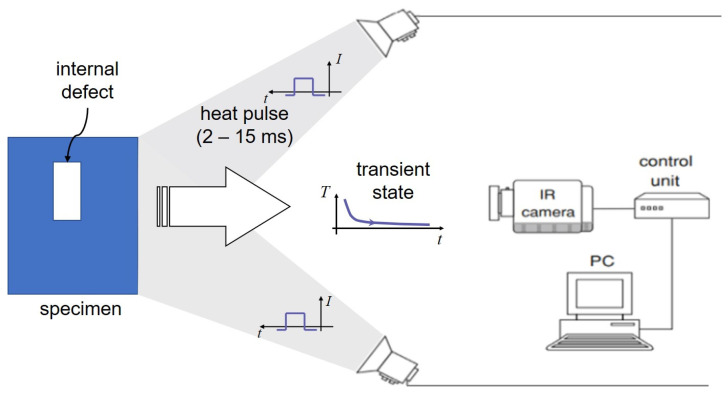
Pulsed thermographic testing using optical excitation.

**Figure 3 sensors-23-04444-f003:**
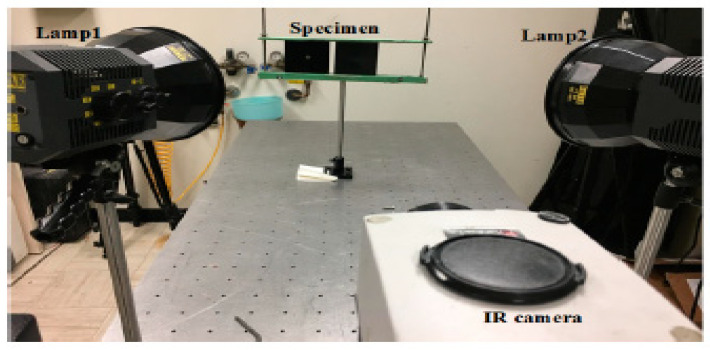
Pulsed thermography experiment platform.

**Figure 4 sensors-23-04444-f004:**
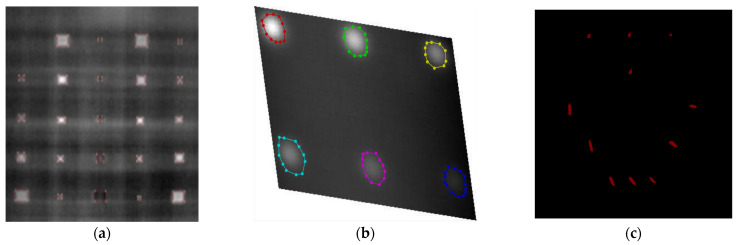
Processing of labelling. (**a**) bounding box labelling; (**b**) circle labelling; (**c**) irregular labelling.

**Figure 5 sensors-23-04444-f005:**
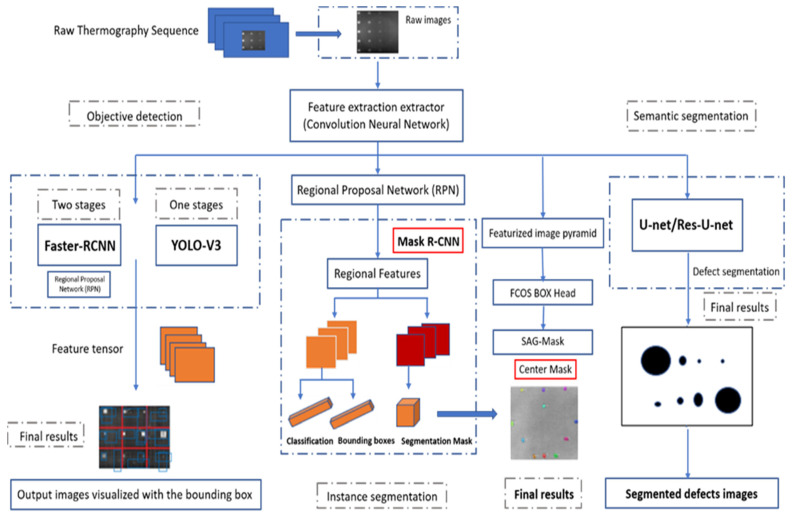
Three types of deep-learning methods (objective detection; instance segmentation; semantic segmentation).

**Figure 6 sensors-23-04444-f006:**
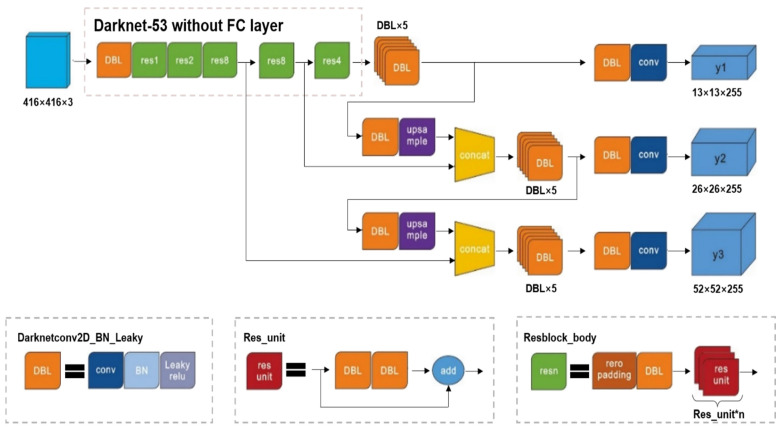
The architecture of residual units in Yolo-v3.

**Figure 7 sensors-23-04444-f007:**
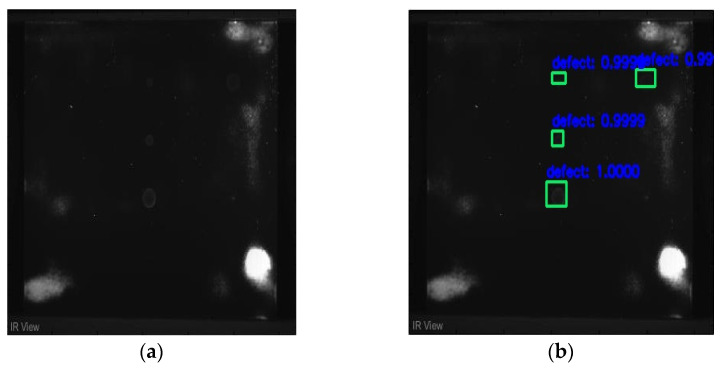
An example of used Method A: (**a**) the original thermal image; (**b**) the detected image.

**Figure 8 sensors-23-04444-f008:**
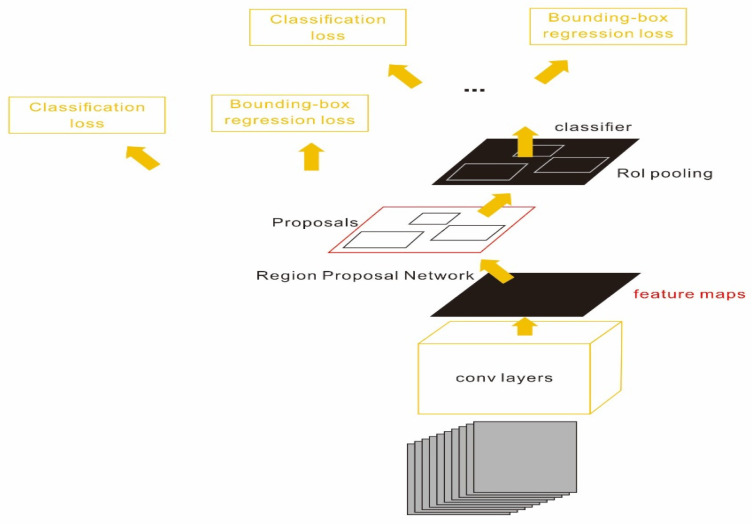
Faster–RCNN defect detection for infrared data.

**Figure 9 sensors-23-04444-f009:**
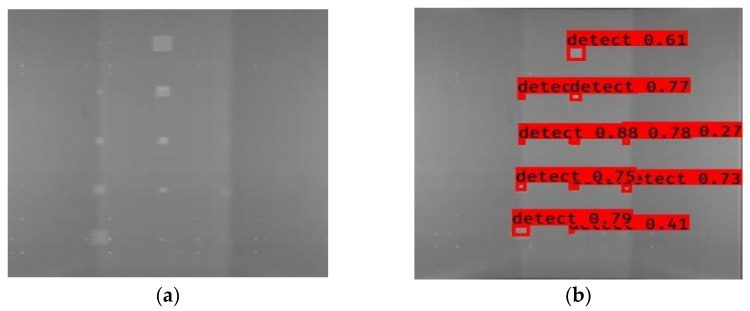
An example of Method B: (**a**) The original thermal image; (**b**) The Faster–RCNN-detected image.

**Figure 10 sensors-23-04444-f010:**
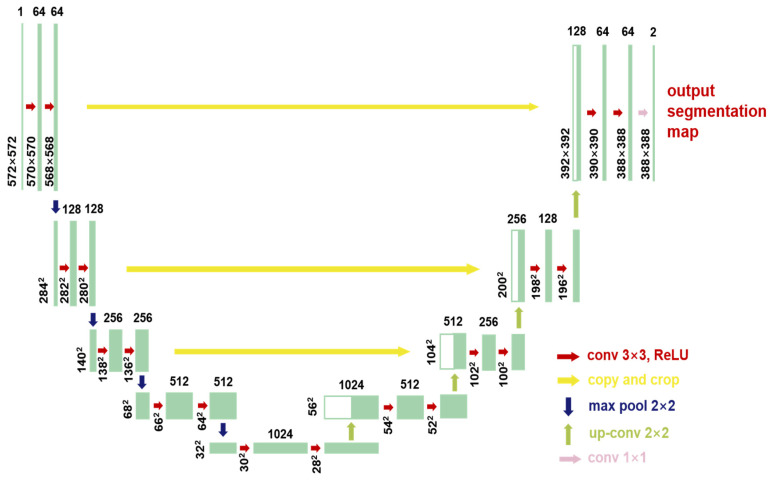
U-net model structure.

**Figure 11 sensors-23-04444-f011:**
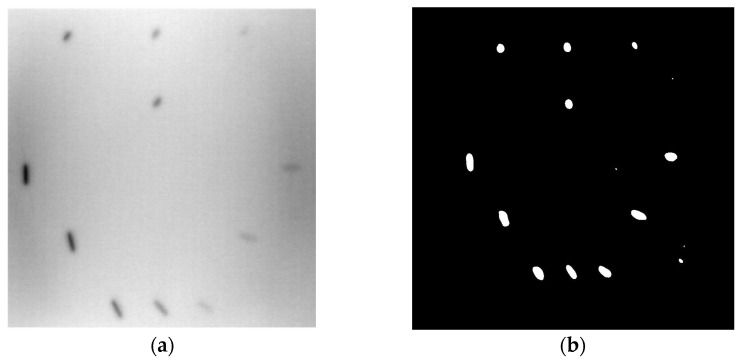
An example of used Method C: (**a**) the original thermal image; (**b**) the being segmented image.

**Figure 12 sensors-23-04444-f012:**
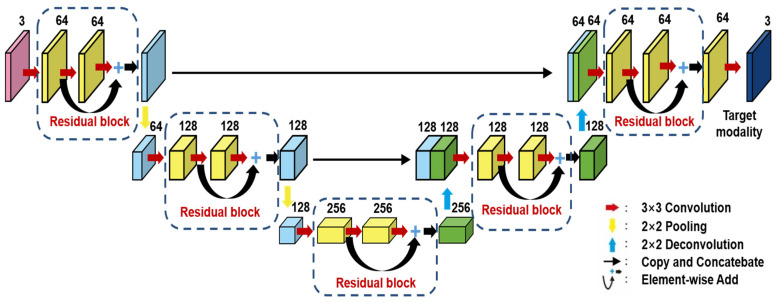
Res–U-net model structure.

**Figure 13 sensors-23-04444-f013:**
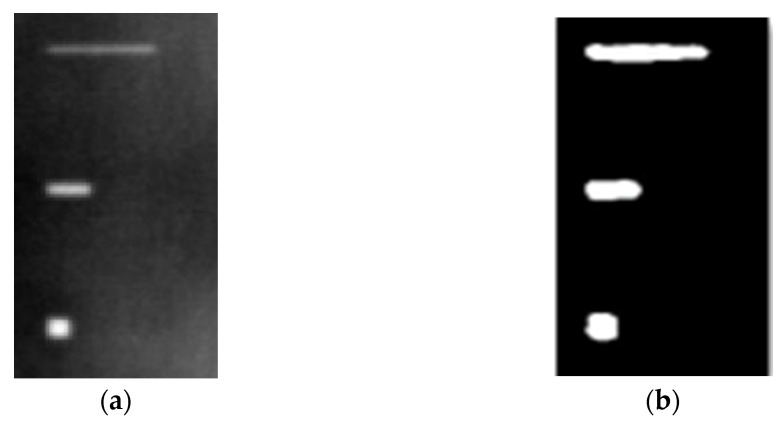
An example of Method 4: (**a**) the original thermal image; (**b**) the segmented image.

**Figure 14 sensors-23-04444-f014:**
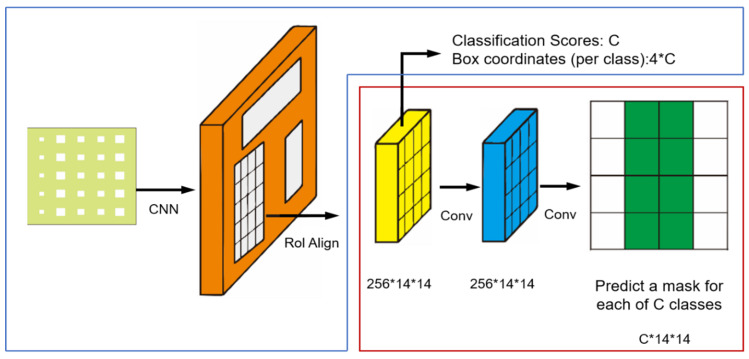
Mask–RCNN processing architecture.

**Figure 15 sensors-23-04444-f015:**
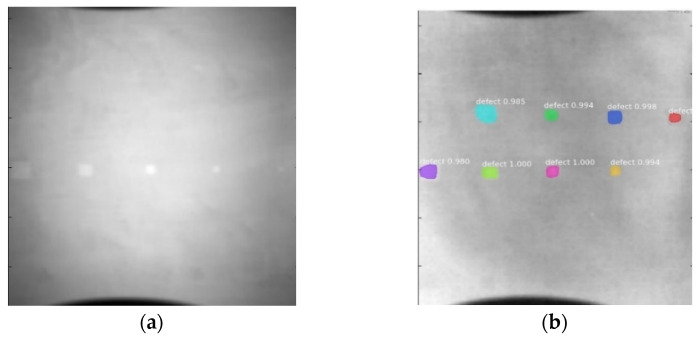
An example of used Method 5: (**a**) the original thermal image; (**b**) the detected image.

**Figure 16 sensors-23-04444-f016:**
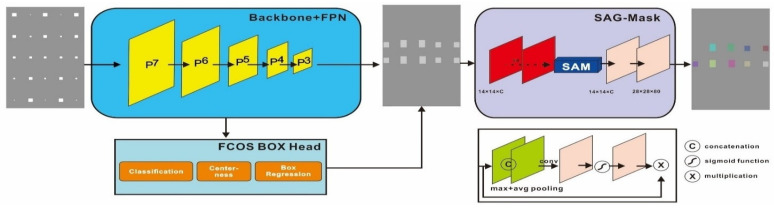
The structure of Center–Mask.

**Figure 17 sensors-23-04444-f017:**
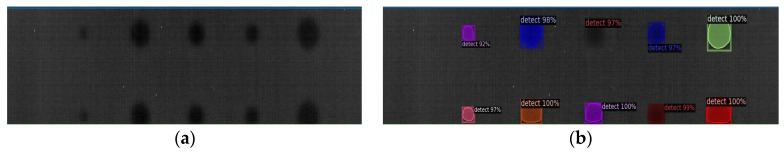
An example of used Method: 6(**a**) the original thermal image; (**b**) the detected image.

**Figure 18 sensors-23-04444-f018:**
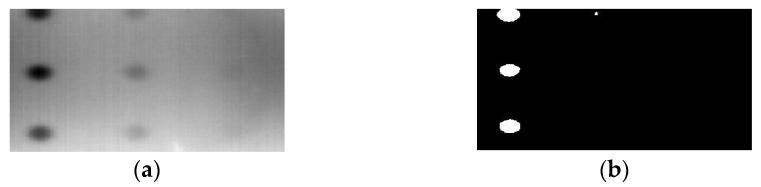
An instance of Method 7 applied on the thermal image (**a**) the original thermal image; (**b**) the detected image.

**Figure 19 sensors-23-04444-f019:**
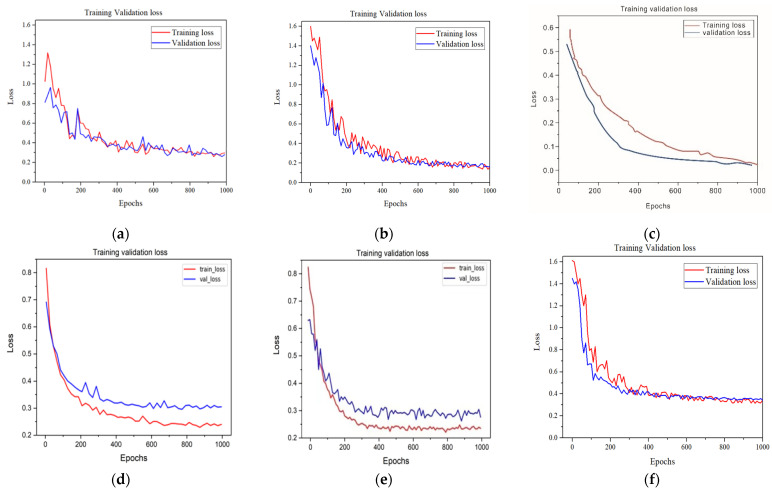
The loss curves in each deep segmentation model (**a**) Mask–RCNN; (**b**) Center–Mask; (**c**) YOLO-V3; (**d**) Res–U-net; (**e**) U-net; (**f**) Faster–RCNN.

**Figure 20 sensors-23-04444-f020:**
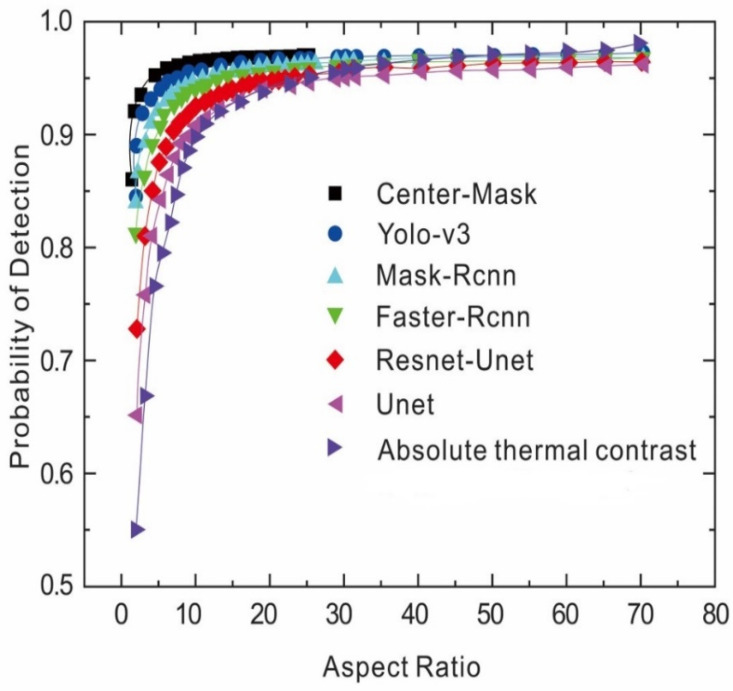
Different detection methods with CFRP sample (CTS = 0.75).

**Figure 21 sensors-23-04444-f021:**
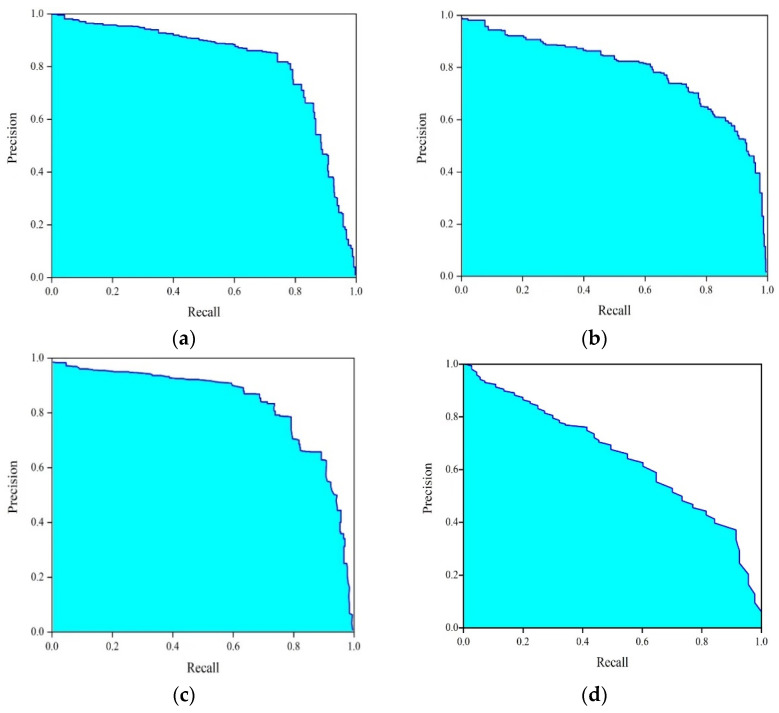
The mean–average precision curve from each deep-segmentation model: (**a**) Mask–RCNN; (**b**) Center–Mask; (**c**) YOLO-V3; (**d**) Faster–RCNN.

**Figure 22 sensors-23-04444-f022:**
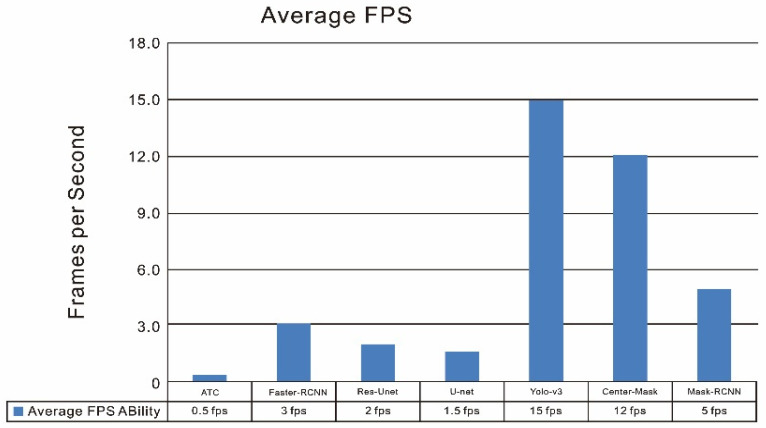
Average frame per second for each deep-learning model.

**Table 1 sensors-23-04444-t001:** The description of each experimental sample.

Number	Type of Materials	Geometrics Specimen	Cross Section	Dimension	Defect Diameters (mm)
1	Plexiglass	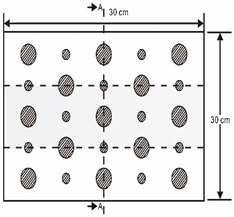	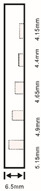	30 cm × 30 cm	Depth: 5.15 mm, 4.9 mm, 4.65 mm, 4.4 mm, 4.15 mm; Diameter: 9 mm, 18 mm
2	Plexiglass	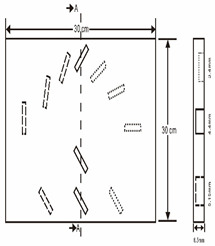	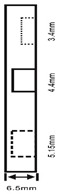	30 cm × 30 cm	Different angle cracks (0°,15°, 30°, 45° 60°, 75°, 90°); Size: 15 mm × 3 mm; Depth: 3.4 mm; 4.4 mm; 5.15 mm;
3	Plexiglass	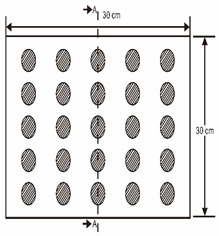	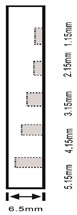	30 cm × 30 cm	Depth: 3.4 mm; 4.4 mm; 5.15 mm; Diameter: 10 mm
4	Plexiglass	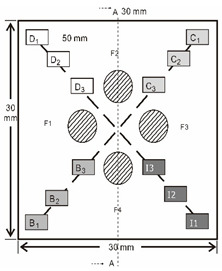	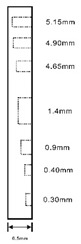	30 cm × 30 cm	Depth: 0.2, 0.4, 0.6, 0.8, 1.0; Diameter or Size: 3.4, 5.6, 7.9, 11.3, 16.9 D1: 15 mm × 6 mm; D2: 8 mm × 5 mm D3: 5 mm × 4 mm; C1: 14 mm × 6 mm C2: 9 mm × 5 mm; C3: 6 mm × 4 mm B1: 16 mm × 6 mm; B2: 15 mm × 6 mm B3: 8 mm × 4 mm; I1: 18 mm × 6 mm I2: 11 mm × 6 mm; I3: 7 mm × 4 mm
5	Steel	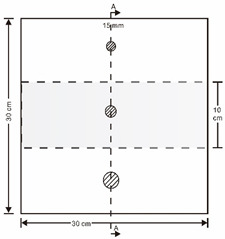	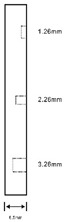	30 cm × 30 cm	Depth: Size: A = 5 mm × 5.0 mm B = 2.5 mm × 10 mm C = 1 mm × 25 mm
6	CFRP	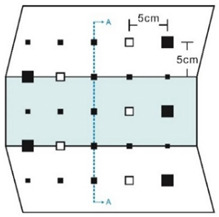	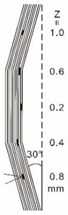	30 cm × 30 cm	Five equivalent diameters (3.4 mm, 5.6 mm, 7.9 mm, 11.3 mm, 16.9 mm) with five different depth of defects (1.0 mm; 0.6 mm; 0.2 mm; 0.4 mm;0.8 mm) The plate has two time folding and at 30 degrees to the horizontal level
7	CFRP	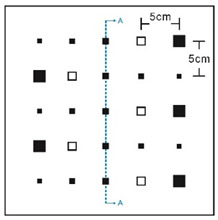	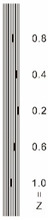	30 cm × 30 cm	Five different lateral size of defects: (3 mm, 5 mm, 7 mm, 10 mm, 15 mm) with five different depth (0.2 mm; 0.4 mm; 0.6 mm; 0.8 mm; 1.0 mm)
8	CFRP	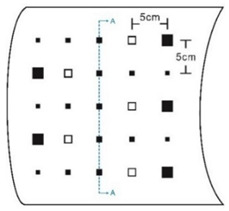	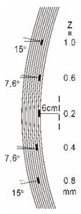	30 cm × 30 cm	Different angle of defects ($0^{{}^{\circ}}$; $7.6^{{}^{\circ}};15^{{}^{\circ}}$) with different equivalent diameter (3.4 mm, 5.6 mm, 7.9 mm, 11.3 mm, 16.9 mm) and corresponding depth (1.0 mm, $15^{{}^{\circ}}$), (0.6 mm, $7.6^{{}^{\circ}}$), (0.2 mm, 0°), (0.4 mm, $7.6^{{}^{\circ}}$), (0.8 mm, $15^{{}^{\circ}}$)

**Table 2 sensors-23-04444-t002:** The training detail from each DL model.

Deep Learning Model	Training Detailed Information	CPU Time
U-net	1. A high-learning momentum (0.99); 2. The weight decay is set as—0.005; 3. A learning rate is 0.005; 4. The number of epochs is 200; 5. The input sequences and the corresponding segmentation images are used to train with U-net with mini-batch gradient descent implantation from Pytorch deep-learning framework.	2 min 12 s
Res–U-net	1. The Res–U-net in this work was trained based on the MXNET [42] deep-learning library; 2. A batch size of 256 with mammal gradient aggregation [43]; 3. The number of epochs is 300; 4. The weight decay is set as—0.00066; 5. Adam optimizer with initial rate 0.005; 6. A multi-dimension learning momentum [44]; $\left(\beta_{1},\beta_{2}\right)$ is (0.9,0,09) and the initial learning rate is 0.0005.	2 min 5 s
YOLO-V3	1. Momentum was implemented as optimizer during the training; 2. The learning momentum was 0.9; 3. The learning rate was set as 0.001; 4. The backbone is adapted Darknet51; 5. The weight decay was 0.0005; 6. The maximum batch size of iteration was 50,200.	45 s
Faster–RCNN	Batch size 256; Overlap threshold for ROI 0.5; Learning rate 0.001; Momentum for SGD 0.9; Weight decay for regularization 0.0001	1 min 55 s
Mask–RCNN	1. Network training using Resnet50 as backbone; 2. The mini mask size is 28 × 28; 3. The weight decay is set as—0.0001; 4. The loss weight is equal for each class and mask ( RPN class, RPN bounding box, MRCNN class, MRCNN bounding box and MRCNN mask); 5. The learning momentum is 0.9 and learning rate is 0.0003; 6. Training of the first 20 epochs of network heads was followed by the training of all network layers for 80 epochs, the model weight.	1 min 45 s
Center–Mask	Stochastic Gradient Descent (SGD) for 90K iterations (200 epoch) with a mini batch of two images and initial learning rate of 0.01; a weight decay of 0.9 and a momentum of 0.01, respectively. All backbone models are initialized by ImageNet pre-trained weights.	1 min 10 s

**Table 3 sensors-23-04444-t003:** Results with semantic segmentation and object localization algorithms.

	Res–U-Net	U-Net	Faster–RCNN	Yolo-v3
(a)	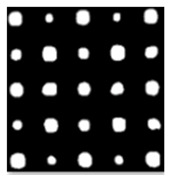	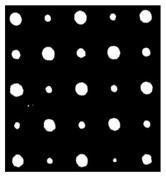	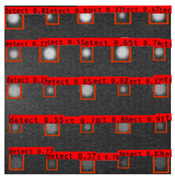	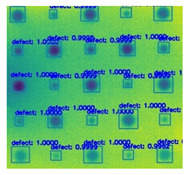
(b)	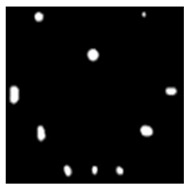	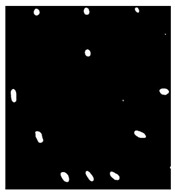	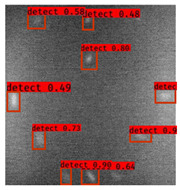	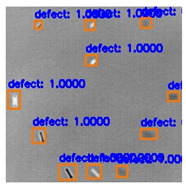
(c)	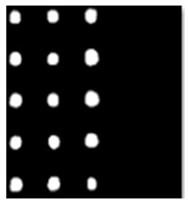	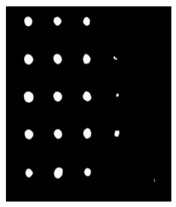	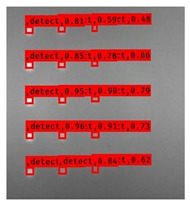	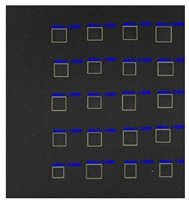
(d)	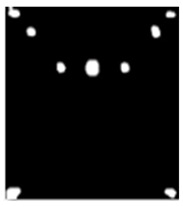	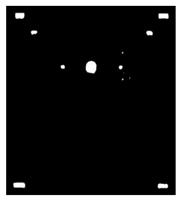	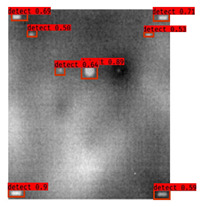	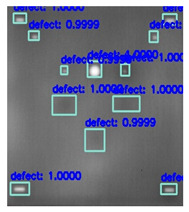
(e)	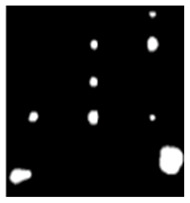	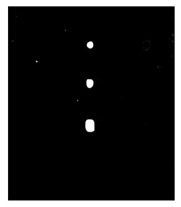	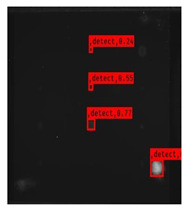	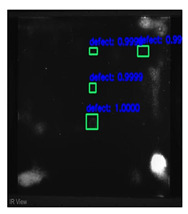
(f)	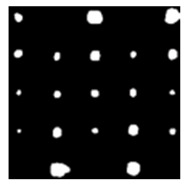	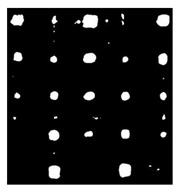	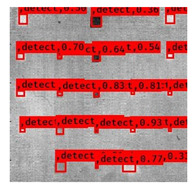	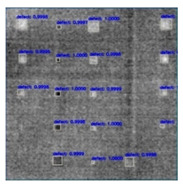
(g)	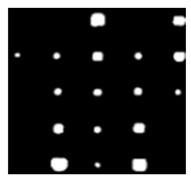	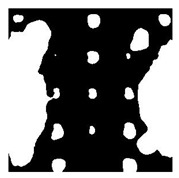	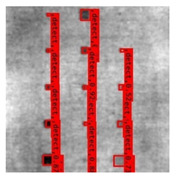	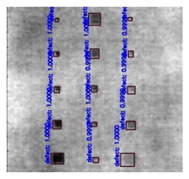
(h)	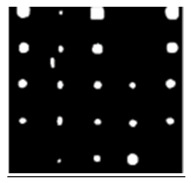	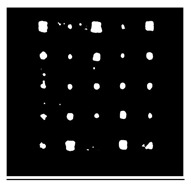	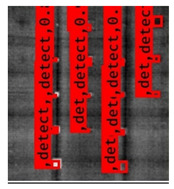	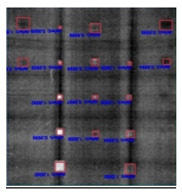

**Table 4 sensors-23-04444-t004:** Segmentation results with instance segmented algorithms; Absolute Thermal Contrast (ATC); Raw.

	Raw	ATC	Mask–RCNN	Center–Mask
(a)	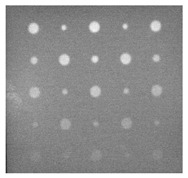	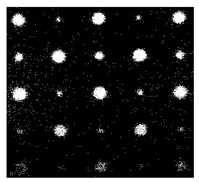	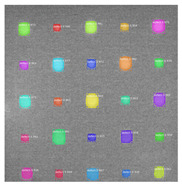	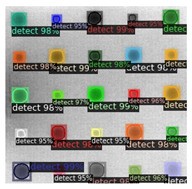
(b)	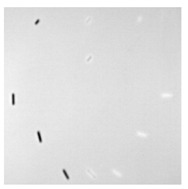	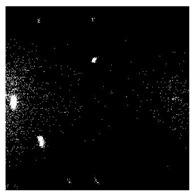	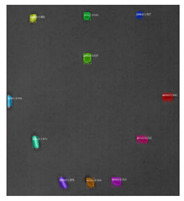	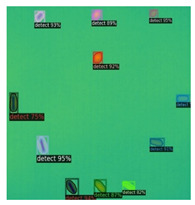
(c)	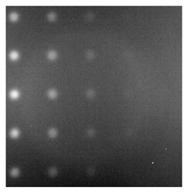	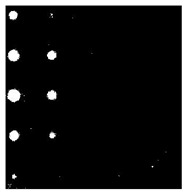	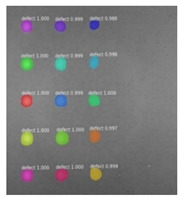	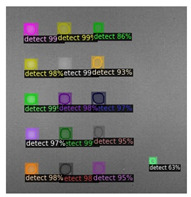
(d)	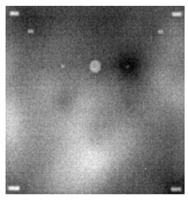	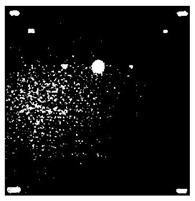		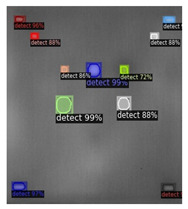
(e)	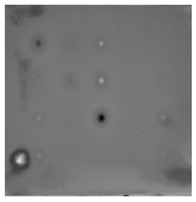	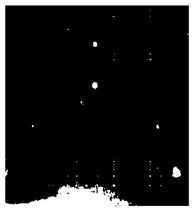	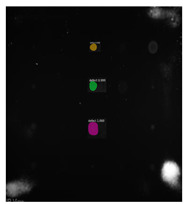	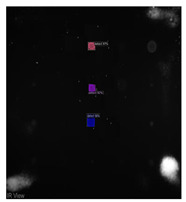
(f)	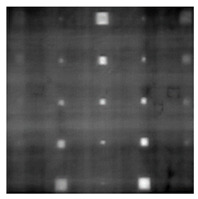	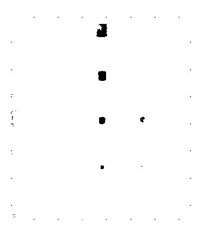	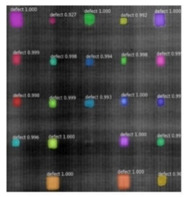	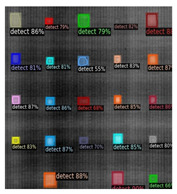
(g)	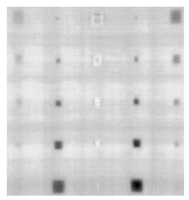	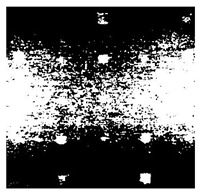	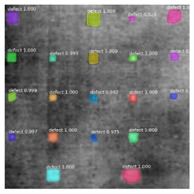	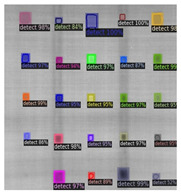
(h)	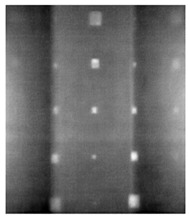	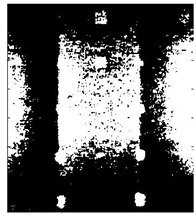	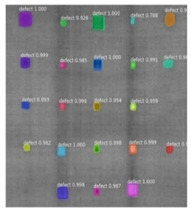	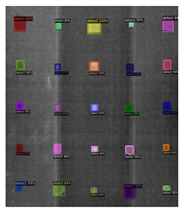

**Table 5 sensors-23-04444-t005:** Different detection methods of results with individual sample (CTS = 0.75).

Methods	POD of Different Samples							
	Sample (a)	Sample (b)	Sample (c)	Sample (d)	Sample (e)	Sample (f)	Sample (g)	Sample (h)
Mask–RCNN	1	0.91	0.87	0.84	0.80	0.91	0.89	0.87
Center–Mask	1	0.94	0.92	0.94	0.82	0.91	0.92	0.95
U-net	0.81	0.84	0.85	0.81	0.75	0.80	0.63	0.76
Res–U-net	0.89	0.87	0.83	0.83	0.79	0.85	0.79	0.85
Faster–RCNN	0.90	0.88	0.84	0.83	0.80	0.85	0.90	0.86
YOLO-V3	1	0.92	0.91	0.90	0.82	0.82	0.94	0.89
Absolute thermal contrast with global threshold	0.65	0.71	0.73	0.78	0.61	0.73	0.57	0.65

**Table 6 sensors-23-04444-t006:** Total detection results with different deep-learning segmentation algorithms (CTS = 0.75).

Samples	Evaluations	Methods						
		Mask–RCNN	U-Net	Res–U-net	Faster–RCNN	Yolo-v3	Center–Mask	ATC
A	Precision	0.46	0.45	0.45	0.47	0.40	0.45	0.30
	Recall	1	0.81	0.89	0.90	1	1	0.75
	F-scores	80%	70%	74%	76%	76%	80%	57.6%
B	Precision	0.41	0.50	0.46	0.43	0.52	0.55	0.25
	Recall	0.91	0.84	0.87	0.88	0.92	0.94	0.69
	F-scores	73.2%	74%	73.8%	72%	80%	82%	51%
C	Precision	0.45	0.44	0.47	0.49	0.57	0.59	0.28
	Recall	0.87	0.85	0.83	0.84	0.91	0.92	0.63
	F-scores	83%	72%	71.9%	73%	81%	82%	50%
D	Precision	0.46	0.47	0.49	0.40	0.59	0.60	0.22
	Recall	0.84	0.81	0.83	0.83	0.90	0.94	0.78
	F-sccores	73.3%	71.6%	72.8%	72%	81.4%	84.4%	52%
E	Precision	0.41	0.40	0.50	0.52	0.60	0.66	0.38
	Recall	0.80	0.75	0.79	0.80	0.82	0.82	0.66
	F-scores	67.2%	64%	70%	72%	76%	78%	57%
F	Precision	0.42	0.35	0.46	0.41	0.64	0.67	0.38
	Recall	0.91	0.80	0.85	0.85	0.82	0.91	0.61
	F-scores	73.7%	57.2%	74.9%	70%	78%	85%	54%
G	Precision	0.49	0.42	0.60	0.41	0.65	0.61	0.46
	Recall	0.89	0.63	0.79	0.90	0.94	0.92	0.69
	F-scores	76.5%	57.2%	74.9%	73%	87%	83%	62%
H	Precision	0.42	0.59	0.55	0.42	0.54	0.64	0.31
	Recall	0.87	0.76	0.85	0.89	0.89	0.95	0.75
	F-scores	71.64%	71.8%	77%	73%	79%	86%	58%
Average	F-scores	74.8%	67.25%	70.66%	72.62%	79.8%	82.55%	55.2%

The precision and recall values always tend to be negatively correlated in the evaluation of DL models. In this project, we placed more emphasis on the recall values and comprehensive F-scores to assess the number of defects, which successfully detected model performance.

## Data Availability

Data available on request from the authors.

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
