# Peer review of "Automatic Detection and Identification of Defects by Deep Learning Algorithms from Pulsed Thermography Data"

_sensors, 2023, doi:10.3390/s23094444_

Round 1
Reviewer 1 Report
Is it possible to integrate any physical process in the IRT to give prominence to this research?
Is this combination of algorithms specific for the IRT?
How do you locate the positions of the defects?
What if the defects are the cracks-like ones? How to change the labeling techniques correspondingly?
Author Response
Dear Professors (reviewer),
Thank you so much for your detailed comments concerning about my manuscript (Sensors-2092863).
Those comments are very helpful and beneficial to revise and improve my paper as well as guide my research in the future. Really appreciate it!
Please see the attachment.

Reviewer 2 Report
The paper proposes an investigation and comparison of three types of deep learning methods in terms of defect detection accuracy and efficiency analysis applied in pulsed thermography. The authors provided a novel instance segmentation method for defect segmentation and identification to process thermal images at the pixel level. Deep learning feature extraction is used for data post-processing.
The topic is interesting and original especially for thermography fault detection method. The paper shows the efficiency of deep learning to extract defect presence in thermography analysis. The references are suitable and appropriate. Some figures should be improved for quality reading.
Some aspects should be further discussed by the authors:
- 1/CPU time for processing should be added and discussed
- 2/if the structure is more complex, how will be the efficiency of the method
Author Response
Dear Professor (reviewer),
Thank you so much for your detailed comments concerning my manuscript (Sensors-2092863).
Those comments are very helpful and beneficial to revise and improve my paper as well as guidance my research in the future. Really appreciate it!
Please see the attachment.

Reviewer 3 Report
1) For the paragraph in Line 90, there are not sufficient references or reviews for infrared thermography defect detection using deep learning.
2) What is the difference between the infrared images obtained from pulsed thermography and other infrared images. It should be described in the introduction. Because it is very important for the deep learning.
3) The contribution 1 and 3 are not lack of innovations.
4) For the experimental samples in Table I, the background of the images obtained seems not complicated. For these images provided, I think the traditional methods can also deal with them. Why do we choose the deep learning methods.
5) All the methods introduced in Section 5 are well-known, there are not innovative structures or adjustment for pulse thermography.
Author Response
Dear Professor (reviewer),
Thank you so much for your detailed comments concerning my manuscript (Sensors-2092863).
Those comments are very helpful and beneficial to revise and improve my paper as well as guide my research in the future. Really appreciate it!
Please see the attachment.

Round 2
Reviewer 3 Report
The author should improve the abstract to make it clear.
The author should improve the quality of all figures in this paper.
The organization of this paper is disordered, a little unclear.
Author Response
Dear Professors (reviewer),
Thank you very much again for providing me with such detailed feedback on my manuscript (Sensors-2092863).
Your comments have been extremely helpful in revising and improving my paper, as well as guiding my future research. I am very grateful for your assistance.
Please see the attachment.
